# Protocol for a definitive randomised controlled trial and economic evaluation of a community-based rehabilitation programme following hip fracture: fracture in the elderly multidisciplinary rehabilitation—phase III (FEMuR III)

Nefyn Williams [1], Susanna Dodd,[2] Ben Hardwick,[2] Dannii Clayton,[2] Rhiannon Tudor Edwards,[3] Joanna Mary Charles,[3] Phillipa Logan,[4] Monica Busse,[5] Ruth Lewis,[6] Toby O Smith,[7] Catherine Sackley,[8] Val Morrison,[9] Andrew Lemmey,[10] Patricia Masterson-Algar [11], Lola Howard,[2] Sophie Hennessy,[2] Claire Soady,[2] Penelope Ralph,[1] Susan Dobson,[1] Shanaz Dorkenoo[12]

For numbered affiliations see end of article.

**Correspondence to**
Professor Nefyn Williams;
nefyn.williams@liverpool.ac.uk

## ABSTRACT

**Introduction** Proximal femoral (hip) fracture is common, serious and costly. Rehabilitation may improve functional recovery but evidence of effectiveness and cost-effectiveness are lacking. An enhanced rehabilitation intervention was previously developed and a feasibility study tested the methods used for this randomised controlled trial (RCT). The objectives are to compare the effectiveness and cost-effectiveness of the enhanced rehabilitation programme following surgical repair of proximal femoral fracture in older people compared with usual care.

**Methods and analysis** Protocol for phase III, parallel-group, two-armed, superiority, pragmatic RCT with 1:1 allocation ratio; allocation sequence by minimisation programme with a built-in random element; secure web-based allocation concealment. The two treatments will be usual care (control) and usual care plus an enhanced rehabilitation programme (intervention). The enhanced rehabilitation will consist of a patient-held information workbook, goal setting diary and up to six additional therapy sessions. Outcome assessment and statistical analysis will be performed blind; patient and carer participants will be unblinded. Outcomes will be measured at baseline, 17 and 52 weeks' follow-up. Primary outcome at 52 weeks will be the Nottingham Extended Activities of Daily Living scale. Secondary outcomes will measure anxiety and depression, health utility, cognitive status, hip pain intensity, falls self-efficacy, fear of falling, grip strength and physical function. Carer strain, anxiety and depression will be measured in carers. All safety events will be recorded, and serious adverse events will be assessed to determine whether they are related to the intervention and expected. Concurrent economic evaluation will be a cost-utility analysis from a health service and personal social care perspective. An embedded process evaluation will determine the mechanisms and processes that explain the implementation and impacts of the enhanced rehabilitation programme.

### Strengths and limitations of this study

► Pragmatic phase III randomised controlled trial following phase I intervention development and phase II feasibility study.
► Concurrent economic evaluation with a health service and personal social care perspective.
► Embedded process evaluation to determine the mechanisms and processes that explain the implementation and impacts of the enhanced rehabilitation programme.
► Only patients with mental capacity to consent are eligible, therefore excluding a large number of potential participants lacking capacity.

**Ethics and dissemination** National Health Service research ethics approval reference 18/NE/0300. Results will be disseminated by peer-reviewed publication.
**Trial registration number** ISRCTN28376407; Pre-results registered on 23 November 2018.

## INTRODUCTION

Proximal femoral fracture, more commonly referred to as hip fracture, is a common, major health problem in old age.[1] It is projected to increase further as the population ages.[2 3] Mortality is high,[4 5] and of those who survive to 1 year, 29% fail to regain their level of functioning, in terms of restrictions of activities of daily living[6]; many lose their independence. This imposes a large cost burden on society, estimated to be approximately £2.3 billion a year in the UK.[7] The majority of costs is incurred in the community and social care setting in the 12 months

following hospital discharge, which are almost four times higher than the costs of the acute hospital admission.[8] Frail individuals are at particular risk of secondary future proximal femoral fracture, resulting in worse morbidity and mortality outcomes.[9]

The National Institute of Health and Clinical Excellence (NICE) have issued guidelines for the management of hip fracture.[10] This includes the provision of a co-ordinated multidisciplinary rehabilitation programme starting in hospital during postoperative recovery and continuing in the community following discharge.[10] Where possible such rehabilitation programmes should consider individual patient goals, facilitate a return to prefracture independence and provide patients and carers with written information to support the rehabilitation programme and long-term outcomes. The Hip Sprint audit reported that community rehabilitation services were inconsistent.[11]

### Rationale

There have been four relevant Cochrane systematic reviews with inconclusive results.[12–15] These have examined different types and intensities of in-patient rehabilitation,[12] mobilisation strategies,[13] psychosocial functioning after hip fracture[14] and rehabilitation for those with dementia following hip fracture surgery.[15] Other systematic reviews have reported improved walking ability,[16] strength and physical function,[17] including those with mild to moderate dementia.[18] These systematic reviews concluded that while individual components of rehabilitation programmes may aid recovery after a hip fracture, there is insufficient evidence to demonstrate clinical effectiveness or cost-effectiveness of an overall care pathway, and that further research is required.

A previous study[19] completed the first two phases of the Medical Research Council framework for complex interventions.[20] The first phase developed an enhanced rehabilitation intervention which, in addition to usual care, included a patient-held workbook, a goal setting diary and up to six additional home-based therapy sessions.[21] The second phase of the study was a randomised feasibility study, which assessed the acceptability of the new rehabilitation programme and the feasibility of trial methods for a definitive phase III randomised controlled trial (RCT).[22 23] Although this feasibility study was underpowered to assess effectiveness, the intervention showed a medium-sized improvement in the Nottingham Extended Activities of Daily Living (NEADL) scale compared with usual care (Cohen's d 0.63). A process evaluation described the implementation of the rehabilitation programme and informed how to enhance recruitment and improve the intervention.[24]

### Risk and benefits

The enhanced rehabilitation programme demonstrated a potential improvement in activities of daily living in the feasibility study. Possible risks of rehabilitation interventions would include injury or falling when performing therapeutic exercises, which must be weighed against the risk to health of sedentary behaviour.

### Primary objective

To determine the effectiveness of an enhanced rehabilitation programme following surgical repair of proximal femoral fracture in older people compared with usual care, in terms of the performance of activities of daily living at 52 weeks follow-up.

### Secondary objectives

1. To compare the cost-effectiveness of an enhanced rehabilitation programme following surgical repair of proximal femoral fracture in older people compared with usual care at 52 weeks follow-up.
2. To determine the effectiveness of an enhanced rehabilitation programme following surgical repair of proximal femoral fracture in older people compared with usual care, in terms of the performance of activities of daily living at 17 weeks follow-up.
3. To determine the effectiveness of an enhanced rehabilitation package following surgical repair of proximal femoral fracture in older people compared with usual care, in terms of anxiety and depression at 17 and 52 weeks follow-up.
4. To assess whether the enhanced rehabilitation intervention creates change in self-efficacy, hip pain, cognitive function, fear of falling and physical function as potential mediators for improving activities of daily living at 17 and 52 weeks follow-up.
5. To assess whether the enhanced rehabilitation intervention creates change in strain, anxiety and depression in carers at 17 and 52 weeks follow-up.
6. To determine the mechanisms and processes that explain the implementation and impacts of the enhanced rehabilitation programme and whether there are adverse effects.

### METHODS AND ANALYSIS
### Trial design

This is a pragmatic, multisite, parallel-group, two-armed, superiority RCT with 1:1 allocation ratio and an internal pilot phase (figure 1). Outcome assessment and statistical analysis will be blinded; patient and carer participants and clinicians will be unblinded. A concurrent economic evaluation will be a cost-utility analysis from a health service and personal social care perspective. An embedded process evaluation will examine the mechanisms and processes that explain the implementation and impacts of the enhanced rehabilitation programme. The RCT was registered on 23 November 2018 . Trial registration data can be found in online supplemental appendix 1.

### Trial setting and selection of sites/clinicians

Sites were recruited by coinvestigators in different regions of England and Wales with a spread of socioeconomic conditions and a mixture of rural and urban locations: Kent (CS), Merseyside (NW), Norwich (TOS), North

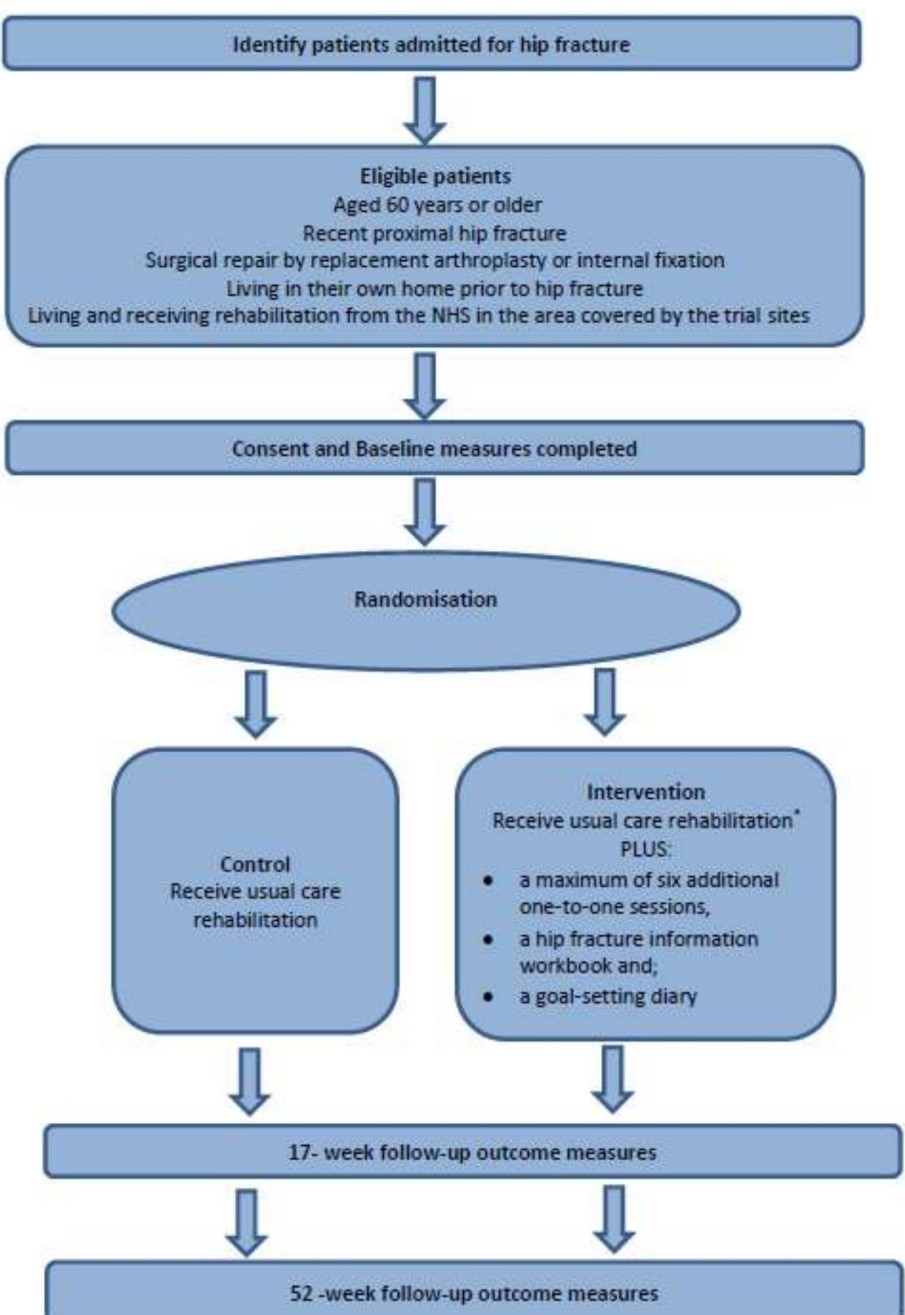

**Figure 1** Participant flowchart for Fracturein the Elderly Multidisciplinary Rehabilitation—phase III (FEMuR III).

Wales (RL), Nottingham (PL) and South Wales (MB). The sites had to include trauma centres treating proximal femoral fracture and links to community rehabilitation teams, which could accommodate the extra community rehabilitation sessions.

Patients will be recruited on orthopaedic, rehabilitation and community hospital wards or after hospital discharge home. The intervention will be delivered in the community, following hospital discharge, by community teams receiving referrals from the acute hospital sites and their associated community hospitals.

## Selection of sites/clinicians
Sites have been opened to recruitment in Nottingham, Norfolk, North Wales, South Wales and East Kent. Further

sites are planned in west Kent, Derby and Cheshire plus others. The site trial teams comprise principal investigators, hospital and community NHS staff, research assistants and support staff from clinical research networks.

## Trial population
### Inclusion criteria
1. Age 60 years or older.
2. Recent proximal femoral fracture.
3. Surgical repair by replacement arthroplasty, hemiarthroplasty or internal fixation.
4. Living in their own home prior to hip fracture.
5. Living and receiving rehabilitation from the NHS in the area covered by the trial sites.

## Exclusion criteria

1. Living in residential or nursing homes prior to hip fracture.
2. Participants unable to understand English or Welsh.
3. Lacking mental capacity to give informed consent.

### Carer participants

We will also recruit carer participants to evaluate carer strain, anxiety and depression. These are defined as a relative or friend providing help with activities of daily living or physical care, at least 4 days a week. Carer participants will provide informed consent but will not receive any trial intervention, so will not undergo eligibility screening or randomisation.

### Trial treatment/interventions

We plan to compare an enhanced rehabilitation intervention with usual rehabilitation care.

### Usual rehabilitation care

Usual care consists of a multidisciplinary rehabilitation delivered by the acute hospital, community hospital and community services depending on patients' individual needs at different times during their recovery and on the availability and accessibility of services in different areas. The multidisciplinary team delivering care and rehabilitation may include: orthopaedic surgeons, orthogeriatricians, nurses, physiotherapists, occupational therapists, dieticians, pharmacists, general practitioners (GPs) and social workers. The settings for care include acute orthopaedic or orthogeriatric wards, rehabilitation units in community hospitals, rehabilitation beds in care homes, the patient's own home and care home settings, all delivered by a variety of community teams in both health and social care services. There will be no restrictions on concomitant medications or treatments.

### Enhanced rehabilitation

The main aim of the intervention is to enhance usual rehabilitation by increasing patients' self-efficacy[25] and increasing the amount and quality of patients' practice of physical exercise and activities of daily living to improve functional outcomes at follow-up. Self-efficacy will be enhanced by means of a patient-held information workbook and a goal setting diary. The workbook will include:

► Information about what has happened to them and what to expect from their recovery.
► Information about NHS, council and voluntary sector services including falls' prevention programmes.
► How to manage their recovery, set goals and monitor progress of their rehabilitation; reduce fear of falling.

In addition to whatever community-based rehabilitation is provided as part of usual care, we will provide up to six additional therapy sessions, once patients are discharged home. These can be delivered by physiotherapists, occupational therapists or their assistants, who have been trained to deliver these extra sessions alongside the workbook, using the diary to set goals and monitor progress. The therapists will tailor these extra sessions, so that the total number of sessions used, the time scale for their delivery and the sessions' content will vary between patients according to need but may include the practice of specific exercises and activities of daily living. Throughout the running of this trial, therapists will receive on-going support via emails, newsletters and refresher events.

### Outcomes

Patient participants will complete outcome measures at baseline, 17 and 52 weeks administered by a research assistant blinded to participant allocation. Follow-up assessments will be completed within participants' homes (tables 1 and 2). The primary outcome will be the difference in NEADL scale[26 27] at 52-week follow-up, between the usual rehabilitation arm and the enhanced rehabilitation arm. At baseline, the patient will be asked to recall the 4 weeks prior to hip fracture and not 4 weeks prior to completing this questionnaire. Secondary outcomes will include the Hospital Anxiety and Depression Scale (HADS),[28] economic measures will be EuroQol, EQ-5D-3L[29] and Client Service Receipt Inventory.[30] A reduced version of this will be used at baseline to reduce participant burden as they recover from hip fracture surgery. Potential mediators of outcome will include a Visual Analogue Scale (VAS) for hip pain intensity,[31] Falls Efficacy Scale—International (self-efficacy)[32 33] and Visual Analogue Scale—Fear of Falling (VASFoF).[34]

The research assistant will assess patient participants' cognitive function at baseline, 17 and 52 weeks using the Abbreviated Mental Test Score.[35] The research assistant will measure physical function at baseline, 17 and 52 weeks using the grip strength test[36–38] and using the Short Physical Performance Battery[39 40] at 17 and 52 weeks.

Carer participants will complete the Caregiver Strain Index[41] and the HADS[28] at baseline, 17 and 52 weeks.

Qualitative interviews will take place with patients and carers after 17 weeks. These will gather data on trial participation and intervention design (see the Process evaluation below).

Routinely collected demographic, clinical and recruitment data will include the numbers of patients who are eligible, willing to be randomised, withdraw after randomisation, complete outcome measurements, also reasons for non-completion, age, gender, hip fracture type, surgery type, comorbid conditions, place of residence prior to admission and place of discharge.

### Sample size calculation

The phase II feasibility study results[23] informed the sample size calculation. The adjusted mean difference in the primary outcome measure (NEADL) between the intervention and control group in the feasibility trial was 3.0. Work completed by Wu *et al*[27] has suggested that the minimum clinically significant difference is 2.4; this has been used within the sample size calculation for this phase III RCT. A two-point score in the NEADL scale would equate to an improvement in function from being independent around the home to being able to use

**Table 1** Outcome measures

| Patient completed measures—primary | Description | Range |
|---|---|---|
| Nottingham Extended Activities of Daily Living scale[26 27] | Activities of daily living (mobility, kitchen, domestic, leisure) with higher score indicating greater independence | (0–66) |
| **Patient completed measures—secondary** | | |
| Hospital Anxiety and Depression Scale (HADS)[28] | Anxiety and depression in patients with physical health problems. Two subscales (0–21) with higher score indicate greater anxiety or depression | (0–21) |
| **Patient completed economic measures** | | |
| EuroQol EQ-5D-3L[29] | Health utility index with five dimensions (mobility, self-care, usual activities, pain/discomfort, anxiety/depression) and three levels to give health states converted to a utility weight. Also Visual Analogue Scale (VAS) for health state today | Health utility weight from 0 (death) to 1.0 (perfect health) also with negative values VAS (0–100) |
| Client Service Receipt Inventory[30] | Use of health and social care services | According to activity |
| **Patient completed process measures (potential mediators of outcomes)** | | |
| Visual Analogue Scale (VAS) for hip pain intensity[31] | VAS of current hip pain intensity | (0–10 cm) |
| Falls Efficacy Scale— International (self-efficacy)[32 33] | How concerned a patient is about falling when performing 16 activities of daily living both inside and outside of the home, rated from 1 (not at all concerned) to 4 (very concerned) | (16–64) |
| VAS-Fear of Falling[34] | VAS with higher scores indicating greater fear of falling | (0–10 cm) |
| **Assessment of cognitive function** | | |
| Abbreviated Mental Test Score[35 36] | Detecting and monitoring cognitive impairment. 10 items with lower scores indicating worse cognitive function | (0–10) |
| **Objective measures of physical function** | | |
| Grip strength[37] | Hand dynamometer | According to metre reading |
| Short Physical Performance Battery[40 41] | Physical function tests assessing lower limb function in terms of balance, gait, strength and endurance. Higher score indicates greater function | (0–12) |
| **Carer completed measure—secondary outcome** | | |
| Caregiver Strain Index[42] | 13 items in the domains: employment, financial, physical, social and time. Positive responses to seven or more items indicate a greater level of strain | (0–13) |
| HADS[28] | Anxiety and depression in carers. Two subscales (0–21) with higher score indicate greater anxiety or depression | (0–21) |

public transport or get in and out of a car. The adjusted mean difference between the groups in NEADL in the randomised feasibility study had an SD of 5.8. In this multisite phase III RCT, a more diverse sample would be expected, so a larger SD would be expected. Parker *et al*[42] used NEADL in a rehabilitation RCT and found an SD of 10. Based on the analysis of covariance (ANCOVA) with alpha of 5% and 90% power to detect a difference of 2.4 (SD=10, $R^2$ of covariate=0.52), 352 patient participants would be required to complete the trial over both treatment groups. When considering the 79% retention rate in the feasibility study,[23] the trial would need to recruit 446 patient participants.

## Recruitment and randomisation
### Screening and consent—patient participants
Patients with proximal femoral fracture will be identified and screened for eligibility, including mental capacity, by clinical staff on orthopaedic or rehabilitation wards. If the patients are eligible, and interested in the trial, the trial team researchers would then recruit patients following the trial's informed consent process. Assessment of eligibility may occur over an extended period, if, for example, the patient is experiencing temporary delirium postsurgery. If, during this period, patients are transferred to rehabilitation wards, community hospitals or discharged home, then assessment will continue in these

**Table 2** Fracturein the Elderly Multidisciplinary Rehabilitation—phase III (FEMuR III) protocol schedule of forms and procedures

| Procedures | Screening | Baseline/randomisation* | Trial intervention† | 17 weeks postrandomisation follow-up | Qualitative interviews | 52 weeks postrandomisation follow-up |
|---|---|---|---|---|---|---|
| Eligibility screening and consent | | | | | | |
| Assessment of eligibility criteria | X | | | | | |
| Written and informed consent (patient/carer) | X | | | | | |
| Confirm consent | | X | X | X | X | X |
| Randomisation | | X | | | | |
| Discharge data | | X | | | | |
| Outcome measurement—patient | | | | | | |
| Nottingham Extended Activities of Daily Living | | X | | X | | X |
| Hospital Anxiety and Depression Scale (HADS) | | X | | X | | X |
| Abbreviated Mental Test Score | | X | | X | | X |
| Visual Analogue Scale (VAS) hip pain intensity | | X | | X | | X |
| Falls Efficacy Scale—International | | X | | X | | X |
| Visual Analogue Score—Fear of Falling | | X | | X | | X |
| EuroQol-5D-3L | | X | | X | | X |
| Client Service Receipt Inventory | | X | | X | | X |
| Grip strength | | X | | X | | X |
| Short Physical Performance Battery | | | | X | | X |
| Outcome measurement—carer | | | | | | |
| Caregiver Strain Index | | X | | X | | X |
| HADS | | X | | X | | X |
| Trial intervention† | | | X | | | |
| Qualitative interviews | | | | | | |
| Re-affirm consent verbally specifically for qualitative phone interview. (patient/carer) | | | | | X | |

Continued

**Table 2** Continued

| Procedures | Screening | Baseline/randomisation* | Trial intervention† | 17 weeks postrandomisation follow-up | Qualitative interviews | 52 weeks postrandomisation follow-up |
|---|---|---|---|---|---|---|
| Qualitative telephone interview | | | | | X | |
| Safety event reporting | | | | | | |
| Monitoring of adverse events | | | X | X | X | X |
| Monitoring of serious adverse events | | | X | X | X | X |

Participant follow-up visits should take place at 17 (±2 weeks) and 52 (±2 weeks) weeks postrandomisation.
*Randomisation and baseline should take place no later than 6 weeks after hip fracture repair surgery.
†If randomised to intervention arm.

alternative locations. These assessments will be recorded in a screening log, including any reasons for ineligibility.

## Informed consent—carer participants

For the purpose of this RCT, carers are defined as either a relative or friend caring for a hip fracture patient, helping them with activities of daily living or physical care on at least 4 days a week. They will be identified and recruited following the trial's informed consent process. Copies of the participant information sheets and informed consent forms are found in online supplemental appendix 1.

## Randomisation procedures

Patient participants who provide informed consent will complete baseline outcome measurements prior to randomisation. Randomisation will take place no later than 6 weeks after hip fracture repair surgery. The randomisation will have an allocation ratio of 1:1 within each stratum and across the trial. Randomisation will use a minimisation programme with a built-in random element using factors that will not be made known to individuals in charge of recruitment to minimise any potential for predicting allocation. Randomisation will be completed by secure web access to the remote randomisation site at the clinical trial unit. The therapists delivering the enhanced rehabilitation intervention will receive an automated email when a participant has been allocated to the intervention group.

## Blinding

This is a pragmatic trial comparing two rehabilitation interventions. It will therefore not be possible to blind participants or their clinicians to treatment group allocation. The research assistants will collect outcome measurements blind to treatment allocation. They will not be informed to which group the patient participants have been allocated and will not be present at any of the therapy sessions. Before any home visits for follow-up assessments, they will ask participants not to reveal their treatment allocation. After the final follow-up assessment, they will complete a perception of allocation form, in

order to monitor the level of blinding achieved for these researchers. Data analysis will be performed blind to treatment allocation.

## Internal pilot

An internal pilot-assessed site recruitment and participant recruitment and retention rates for the 6 months after the first site were open to recruitment from September 2019 to February 2020.

## Progression criteria

► Number of sites open: seven or more (go); five to six (amend); four or fewer (stop).
► Open site recruitment rate per month: two or more (go); one to two (amend); less than 1 (stop).
► Retention rate: 69% or higher (go); 50%–68% (amend); 49% or fewer (stop).

## Statistical analysis

Final analysis will take place once all participants have been followed up for 52 weeks, and the database has been locked. Analyses will be by 'intention to treat' for the primary and secondary outcomes on all randomised participants, in the group to which they were allocated and for whom the outcomes of interest have been observed or measured.

### Baseline

Demographic and baseline characteristics will be summarised separately using descriptive statistics for each randomised group to allow clinical assessment of whether balance was achieved between randomised groups. No statistical testing of differences between groups will be performed.

### Analysis of effectiveness

Primary and secondary outcomes at baseline, 17 weeks' and 52 weeks' follow-up will be summarised for each treatment group using descriptive statistics at each time point. If normally distributed, the difference between group means (with 95% CIs) will be reported from the

ANCOVA adjusted for baseline score and stratification factors.

## Missing data and withdrawals

Predictors of missing data will be investigated using regression models (including type of surgery, age, living arrangements and comorbidities) and any significant predictors will be considered for inclusion in the models. In addition, given the two assessment points at 17 and 52 weeks, we will carry out a sensitivity analysis using a joint modelling approach to check whether there is any difference in outcome (here the longitudinal outcome rather than the outcome at 17 weeks or 52 weeks alone) between the randomised arms adjusted for dropouts or missing values.

## Instrumental variable regression

The impact of engagement with the intervention will be assessed using instrumental variable (IV) regression, using the number of face-to-face direct rehabilitation sessions over 52 weeks' follow-up as a continuous measure of engagement. Additional exploratory IV regression analyses will use in turn: the total number of rehabilitation sessions (face-to-face plus telephone), total time (in minutes) spent in face-to-face direct rehabilitation sessions and total time (in minutes) spent in all rehabilitation sessions (ie, face-to-face and telephone). The suitability of using randomisation as the instrument in these IV regression models will be assessed using tests of exogeneity, redundancy and under/weak identification.

## Mediation analyses

The hypothesised mechanism of change for the enhanced rehabilitation intervention is that participants' primary outcome (activities of daily living) is mediated by self-efficacy, hip pain, cognitive function, fear of falling and physical function. If the enhanced rehabilitation intervention has a significant effect on primary outcome ($p<0.05$) for enhanced rehabilitation in ANCOVA, causal mediation analysis will be used to determine whether each of these potential mediators predict change in NEADL at 52 weeks. Initial assessments will determine whether the randomised intervention affects each putative mediator in turn. For those putative mediators that are significantly ($p<0.1$) affected by the randomised intervention, mediation analysis will be carried out adjusting for baseline covariates that predict both the mediator and NEADL, potentially including type of surgery, age, living arrangements (alone/with others) and comorbidities. Sensitivity analyses will assess the potential impact of unmeasured confounding between the mediator and NEADL.

## Economic analysis

The enhanced rehabilitation programme will be fully costed using unit costs from a public sector multiagency perspective. Unit costs will be obtained from national sources of reference costs[43 44] and applied to information received from pilot questionnaires, namely, salary band of therapists, time spent with the patient conducting rehabilitation, costs of travel and costs of any additional equipment. Costs of health and social care services used by the participants will also be costed using national sources of reference costs. The costs of service use and the cost of the intervention will be added together for use in a cost-effectiveness analysis.

The EQ-5D (3L) will be used to calculate Quality Adjusted Life Years (QALYs) over the 52-week trial period, using the area under the curve method.[45 46] A cost-utility analysis will be conducted to calculate a cost per QALY of the enhanced rehabilitation intervention. This cost per QALY generated will be compared with the NICE threshold range of £30 000 per QALY.[47] We will bootstrap differences in costs and outcomes (EQ-5D-3L) between the two groups, producing a 95% CI around these differences.

## Process evaluation

The process evaluation will aim to identify and explain all mechanisms and processes (ie, the intervention theory) that enabled or acted as a barrier to the implementation of the enhanced rehabilitation intervention. The process evaluation will help build a picture of how the intervention was carried out in reality and what factors shaped it. By carrying out a process evaluation, it will be possible to identify if observed impacts are solely due to the enhanced rehabilitation programme, or if these impacts are a result of a number of external and internal variables that are closely linked to the environment and the context in which the intervention takes place.[48–51]

The specific objectives will be to:
► Refine the programme theory from the previous realist review that was used to develop the intervention.[21] This programme theory will explain how the researchers envisage the intervention to work, to reach its expected outcomes.
► Investigate therapists' expectations and experience of implementation, their previous experience and training and their learning throughout the conduct of the trial.
► Investigate the mechanisms driving and shaping the tailoring of the enhanced rehabilitation intervention to individual patients.
► Investigate trial participants' (patients and carers) experiences and views about their involvement in the trial as well as their experience of care in either arm of the trial.
► Map and synthesise all data collected in order to test the refined programme theory and explain the trial findings.

### Process evaluation data collection

Semistructured telephone interviews will be conducted with:
► A purposive sample of 60 patient participants in each of the two trial arms and up to 30 of their carers. Patients will be purposively sampled to ensure diversity based on age, functional impairment (using

baseline NEADL scores) and the presence or absence of a family carer. Interviews will take place after the 17-week assessment and will be audio recorded.

► The therapists delivering the enhanced rehabilitation programme, which will explore implementation from the therapists' perspectives. Interviews will be conducted midway through their involvement in the trial, and at the end, in order to investigate learning over time.

### Data on intervention delivery and adherence

► Therapists will record key reflections on 'critical incident reports'.

► The visiting therapist will record the length and content of each extra rehabilitation therapy session on a case report form.

► All patient participants will be given a therapy session record, where visiting therapists will record the number, length and content of usual rehabilitation care. Whenever possible, routinely collected electronic data that therapists complete on their Therapy Manager system, or its equivalent, will be collected.

► An online questionnaire will be emailed to participating therapists in order to capture therapists' relevant training, previous experience and familiarity with the trial intervention.

Qualitative data will be analysed following a thematic analysis approach[52] that will be guided by the proposed programme theory. Quantitative data (record forms and online questionnaires) will be analysed using descriptive statistics, which will allow the exploration of frequency of responses. All data sets will be synthesised in order to describe the complex nature of the enhanced rehabilitation intervention.

### Patient and public involvement

There has been patient and public involvement (PPI) at all stages including refining the research question, choosing outcomes relevant to patients, commenting on the burden of the intervention and of trial participation. A PPI coinvestigator will continue to contribute to the trial management group, including comments on patient-facing materials and the dissemination plan.

### Ethics and dissemination

NHS research ethics approval was obtained from North East—Tyne & Wear South Research Ethics Committee, reference 18/NE/0300. The current protocol is V.4.0 (11 December 2019). A Trial Steering Committee is providing overall supervision and an Independent Data Safety and Monitoring Committee is responsible for reviewing and assessing recruitment, interim monitoring of safety and effectiveness, trial conduct and external data.

All safety events will be recorded by researchers when they are made aware of the event by the patient, carer, the treating clinicians or therapists. Adverse event reports and serious adverse events (SAEs) not related

to the intervention will be entered on to the remote data entry system. Each SAE will be assessed by the relevant principal investigator (PI) to determine whether it is related to the intervention. A related SAE will be assessed by the CI to determine whether it is expected. If the SAE is related and unexpected, it will be reported to the Research Ethics Committee and sponsor in an expedited manner.

Reporting of the trial will be consistent with the Consolidated Standards of Reporting Trials 2010 Statement (patient-reported outcomes and non-pharmacological interventions).[53] We will submit the final report to a peer-reviewed academic journal, according to our publication strategy and authorship policy. Research data will be available for secondary analysis on reasonable request.

### Trial status

At the time of submission, this trial had been open in nine sites and had recruited 96 patients and 10 carers, with a recruitment rate of two patient participants per site per month and a retention rate of 83%, which fulfilled the progression criteria of the internal pilot. However, recruitment to the trial is currently suspended because of the COVID-19 pandemic. Wherever possible, participants already recruited into the trial will complete their follow-up assessments over the telephone or by post, extra rehabilitation sessions will be delivered over the telephone. When trial recruitment resumes, updated recruitment information will be found on the website http://femur3study.co.uk/

**Author affiliations**
[1]Department of Primary Care and Mental Health, University of Liverpool Faculty of Health and Life Sciences, Liverpool, UK
[2]Liverpool Clinical Trials Centre, University of Liverpool Faculty of Health and Life Sciences, Liverpool, UK
[3]Centre for Health Economics & Medicines Evaluation, Bangor University College of Human Sciences, Bangor, UK
[4]Division of Rehabilitation and Ageing, University of Nottingham Faculty of Medicine and Health Sciences, Nottingham, UK
[5]Centre for Trials Research, Cardiff University, Cardiff, UK
[6]North Wales Centre for Primary Care Research, Bangor University College of Human Sciences, Bangor, UK
[7]School of Health Sciences, University of East Anglia Faculty of Medicine and Health Sciences, Norwich, UK
[8]University of Nottingham Faculty of Medicine and Health Sciences, Nottingham, UK
[9]School of Psychology, Bangor University College of Human Sciences, Bangor, UK
[10]School of Sports, Health and Exercise Science, Bangor University College of Human Sciences, Bangor, UK
[11]School of Healthcare Sciences, Bangor University College of Human Sciences, Bangor, UK
[12]Involving People Network, Health and Care Research Wales, Cardiff, UK

**Acknowledgements** The authors would like to thank all of the principal investigators, research nurses, hospital staff, community rehabilitation teams, physiotherapists, occupational therapists and their assistants who are providing the additional therapy sessions. In Nottingham: Miriam Golding-Day, Katie Robinson, Jane Horne, Frances Allen. In North Wales: Matthew Jones, Lauren Porter, Jane Stockport, Mary Roberts, Victoria Whitehead, Yogesh Joshi, Gareth Evans, Cathy Baker, Jeannie Bishop, Jane Heron, Llinos Davies. In South Wales: Rachel Lowe,

Rebecca Milton, Andrew Dean Young, Peter Lewis, Lisa Roche. In East Anglia: Stephanie Tuck, Catherine Mingay, Stephanie Howard. In East Kent: Danielle Pearce, Joanne Jackson, Sarah Chapman, Andrew Smith, Alex Chipperfield, Joanne Deery. Special thanks to the LCTC data management team Clare Jackson and Dianne Wheatley, LCTC information systems team Keith Kennedy and Janet Harrison, trial coordinator assistant Kieran Crabtree and quality assurance manager Katie Neville. Research assistants/research project support officers who perform follow-up visits. Others who have contributed to the trial management group Kevin Anthony, Gail Arnold, Janine Bates, Suzannah Evans, Mark Howells, Lara Lavelle-Langham, Nic Nikolic, Jessica Roberts. Also the North West Coast Clinical Research Network Lisa Cheng, Julia Murgarza. We very much appreciate the work of the Trial Steering Committee Gail Mountain (chair), Graeme Holt, Helen Hughes, George Kernohan, Tosan Okoro, Nikki Totton and the Independent Data and Safety Monitoring Committee Chris Robertson (chair), Diane Dixon, Peter Giannoudis, Barbara Hanratty.

**Contributors** NW was the chief investigator and grant holder, was responsible for study design, conduct and analysis and had overall responsibility for the study and acts as guarantor. LH and SH were the trial coordinators and BH was the senior trial coordinator overseeing day-to-day conduct, and provided methodological input. LH was the initial trial coordinator and contributed to writing the trial protocol and setting up the trial. SD wrote the statistical analysis plan. RTE and JMC wrote the health economic analysis plan. PM-A wrote the process evaluation analysis plan with NW. RL is a lead applicant from Bangor University and oversees sites in North Wales. MB is a lead applicant from Cardiff University and oversees sites in South Wales. PL is a lead applicant from University of Nottingham and oversees sites in Nottingham. CS is a lead applicant from Kings College London and oversees sites in Kent. TOS is a lead applicant from the University of East Anglia and oversees the site in East Anglia. VM and AL were coinvestigators responsible for study design, methodological oversight and provided health psychology and exercise science expertise respectively. All authors were involved in drafting, revising and approving this manuscript.

**Funding** This work was supported by the National Institute for Health Research's Health Technology Assessment Programme, grant number 16/167/09. The views and opinions expressed therein are those of the authors and do not necessarily reflect those of the HTA, NIHR, NHS or the Department of Health. University of Liverpool: sponsor's reference: UoL001378. Contact: Mr Alex Astor, University of Liverpool, 2nd Floor Block D Waterhouse Building, 3 Brownlow Street, Liverpool L69 3GL. The university has appropriate clinical trials and professional indemnity insurance.

**Competing interests** NHW reports additional grants from NIHR HS&DR outside the submitted work and membership of the NIHR HTA programme funding committee (commissioned research).

**Patient and public involvement** Patients and/or the public were involved in the design, or conduct, or reporting, or dissemination plans of this research. Refer to the Methods section for further details.

**Patient consent for publication** Not required.

**Provenance and peer review** Not commissioned; externally peer reviewed.

**ORCID iDs**
Nefyn Williams http://orcid.org/0000-0002-8078-409X
Patricia Masterson-Algar http://orcid.org/0000-0001-6344-1346

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
