## [Reviewer comments · BMJ Open]

ARTICLE DETAILS

TITLE (PROVISIONAL)	Protocol for a definitive randomised controlled trial and economic evaluation of a community-based rehabilitation programme following hip fracture: Fracture in the Elderly Multidisciplinary Rehabilitation - Phase III (FEMuR III) [ISRCTN28376407]
AUTHORS	Williams, Nefyn; Dodd, Susanna; Hardwick, Ben; Clayton, Dannii; Edwards, Rhiannon; Charles, Joanna; Logan, Phillipa; Busse, Monica; Lewis, Ruth; Smith, Toby; Sackley, Catherine; Morrison, Val; Lemmey, Andrew; Masterson-Algar, Patricia; Howard, Lola; Hennessy, Sophie; Soady, Claire; Ralph, Penelope; Dobson, Susan; Dorkenoo, Shanaz

VERSION 1 - REVIEW

REVIEWER	Howe Tet Sen Department of Orthopaedic Surgery Singapore General Hospital
REVIEW RETURNED	15-May-2020

GENERAL COMMENTS	Suggest that you do randomization of hip fractures that were replaced with a total hip or hemiarthroplasty and hips that were internally fixed separately. Alternatively a sub-group analysis post hoc. I would prefer the former.
--

REVIEWER	Wing-Hoi Cheung The Chinese University of Hong Kong, Hong Kong SAR
REVIEW RETURNED	01-Jun-2020

GENERAL COMMENTS	This is a protocol article for a Phase III randomized controlled trial of evaluating the effectiveness of community-based multidisciplinary rehabilitation programme for hip fracture patients. The study is well designed with appropriate methods. The sample size calculation is sufficiently justified with the findings of previous studies, while statistical analysis is comprehensively planned as well. Randomization and blinding quality are explained adequately. Hence, the protocol is very well written. I have the following minor concerns only: • Under "Selection of sites/clinicians", the authors did not explain any criteria of the site selection, e.g. size of sites, patient load of sites, etc.? Any justification?• Under "Enhanced rehabilitation", it mentions that six additional
--

	therapy will be provided. Could you briefly elaborate what kinds of additional therapy? • Baseline, 17 and 52 weeks are selected as time points of this study. Could you justify why 17 and 52 weeks are selected?
--	---

VERSION 1 – AUTHOR RESPONSE

Reviewer: 1

Suggest that you do randomization of hip fractures that were replaced with a total hip or hemiarthroplasty and hips that were internally fixed separately. Alternatively a sub-group analysis post hoc. I would prefer the former.

REPLY:

Thank you for the suggestion. Ninety six patients have already been randomised in a group which has combined arthroplasty patients with those receiving hemi-arthroplasty or internal fixation, so separate randomisation would not be feasible. We have carefully considered whether to do specific sub-group analyses in the statistical analysis plan. Such analyses are typically subject to low power, which will be a particular problem with the small numbers of participants we anticipate to be recruited with internal fixation. On balance, we do not feel that a sub-group analysis according to type of surgical treatment is justified.

Reviewer: 2

This is a protocol article for a Phase III randomized controlled trial of evaluating the effectiveness of community-based multidisciplinary rehabilitation programme for hip fracture patients. The study is well designed with appropriate methods. The sample size calculation is sufficiently justified with the findings of previous studies, while statistical analysis is comprehensively planned as well. Randomization and blinding quality are explained adequately. Hence, the protocol is very well written.

REPLY: Thank you

I have the following minor concerns only:

- Under “Selection of sites/clinicians”, the authors did not explain any criteria of the site selection, e.g. size of sites, patient load of sites, etc.? Any justification?

REPLY:

The following paragraph has been added to this section; “Sites were recruited by co-investigators in different regions of England and Wales with a spread of socio-economic conditions and a mixture of rural and urban locations: Kent (CS), Merseyside (NW), Norwich (TS), North Wales (RL), Nottingham (PL) and South Wales (MB). The sites had to include trauma centres treating proximal femoral fracture and links to community rehabilitation teams, which could accommodate the extra community rehabilitation sessions.”

- Under “Enhanced rehabilitation”, it mentions that six additional therapy will be provided. Could you briefly elaborate what kinds of additional therapy?

REPLY:

This section has been amended as follows; “In addition to whatever community-based rehabilitation is provided as part of usual care, we will provide up to six additional therapy sessions, once patients are discharged home. These can be delivered by physiotherapists, occupational therapists or their assistants, who have been trained to deliver these extra sessions alongside the workbook, using the

diary to set goals and monitor progress. The therapists will tailor these extra sessions, so that the total number of sessions used, the time scale for their delivery, and the sessions' content will vary between patients according to need, but may include the practice of specific exercises and activities of daily living.”

- Baseline, 17 and 52 weeks are selected as time points of this study. Could you justify why 17 and 52 weeks are selected?

REPLY:

The final follow-up is 12 months (52 weeks) after randomisation as specified in the funder's commissioning brief. The interim follow-up at 4 months (17 weeks) was chosen to mirror the 4-month data collection point for the National Hip Fracture Database in the UK.

VERSION 2 – REVIEW

REVIEWER	Howe Tet Sen Dept of Orthopaedics Singapore General Hospital Singapore
REVIEW RETURNED	12-Jun-2020

GENERAL COMMENTS	Adequate study, accept
------------------------